# OCNet-Based Water Body Extraction from Remote Sensing Images

**Yijie Weng [1], Zongmei Li [1,2],*, Guofeng Tang [1] and Yang Wang [3]**

1   School of Computer and Information Engineering, Xiamen University of Technology, Xiamen 361024, China; 2122031221@s.xmut.edu.cn (Y.W.); 2122031218@s.xmut.edu.cn (G.T.)
2   Institute of Remote Sensing and Digital Earth, Chinese Academy of Sciences, Beijing 100094, China
3   College of Geography and Planning, Nanning Normal University, Nanning 530100, China; wangy.10b@igsnrr.ac.cn
*   Correspondence: lizongmei@xmut.edu.cn; Tel.: +86-13616077596

**Abstract:** Water body extraction techniques from remotely sensed images are crucial in water resources distribution studies, climate change studies and other work. The traditional remote sensing water body extraction has the problems of low accuracy and being time-consuming and laborious, and the water body recognition technique based on deep learning is more efficient and accurate than the traditional threshold method; however, there is the problem that the basic model of semantic segmentation is not well-adapted to complex remote sensing images. Based on this, this study adopts an OCNet feature extraction network to modify the base model of semantic segmentation, and the resulting model achieves excellent performance on water body remote sensing images. Compared with the traditional water body extraction method and the base network, the OCNet modified model has obvious improvement, and is applicable to the extraction of water bodies in true-color remote sensing images such as high-score images and unmanned aerial vehicle remote sensing images. The results show that the model in this study can realize automatic and fast extraction of water bodies from remote sensing images, and the predicted water body image accuracy (ACC) can reach 85%. This study can realize fast and accurate extraction of water bodies, which is of great significance for water resources acquisition and flood disaster prediction.

**Keywords:** water body extraction; remote sensing image; semantic segmentation; OCNet; deep learning

## 1. Introduction

Water is the largest resource on the Earth's surface, occupying 71% of the surface area, with 97.2% of the water distributed in the oceans, 2.15% in icebergs and glaciers, 0.31% in groundwater, 0.009% in lakes, 0.001% in atmospheric water vapor, and 0.0001% in rivers and streams. The groundwater, lakes, rivers and streams available to humans, animals, and vegetation make up less than 1% of the total water on earth. China's freshwater resources total 2800 billion cubic meters, accounting for 6% of global water resources, ranking fourth in the world after Brazil, Russia and Canada. However, China's per capita water resources are only 2300 cubic meters, only one fourth of the world average, making it one of the poorest countries in the world according to per capita water resources. With the continuous development of science and technology, using high spatial-, spectral- and temporal-resolution remote sensing data to realize rapid, accurate and large-scale extraction of water body information, while establishing good applicability to serve all walks of life in people's production, has become the development trend of the water body information extraction research [1]. Water body extraction technology is a key technology in China and even globally, which is indispensable to the survival of human beings and the development of the environment.

Under the continuous progress of satellite remote sensing technology, satellites provide an growing image database, such as remote sensing data from the Landsat [2] series of

satellites, the Sentinel [3] series of satellites, the Gaofen series of satellites and other remote sensing data for lakes, oceans, rivers and other bodies of water, whose analysis provides a large amount of data. However, with the increase of the remote sensing image data volume and spatial resolution, coupled with the high complexity of the Earth's surface environment [4], the current surface water extraction methods are limited in applicability and accuracy [5]. Existing water body data extraction methods are not effective enough in complex images with multiple bands and need to be improved accordingly.

The traditional water body extraction methods are mainly the single-band threshold method [6] and spectral water index method [7–9]. The water body extraction model is constructed by combining the original bands of the image or the characteristic bands obtained by preprocessing [10]. The single-band threshold method only uses image data in near-infrared or infrared bands for analysis, which has some limitations for multi-band remote sensing images. The spectral water index method is applicable to multi-band remote sensing images, generating index data through data of different bands, and then setting the corresponding threshold for water body extraction results. The spectral water index method is also in continuous improvement, such as the normalized water difference index (NDWI) method [11], in which most of the water bodies can be proposed, and the boundary effect is relatively good, but due to the influence of shadows in the image (e.g., algal water bodies), resulting in the water body and the surrounding land values being similar to each other, there is still a small patch and the phenomenon of mispropagation, making it necessary to continue to remove small patches, and to set up according to one's needs the minimum upper map area, thus improving this result. The improved normalized water difference index method (MNDWI) [12] is different from NDWI in that it replaces near-infrared band (NIR) data with short-infrared band (SWIR) data, which significantly reduces the extraction error of the shadow effect, and is conducive to the segmentation of the boundary of the water body; there is not a big difference in the extraction results obtained by using the NDWI and the MNDWI for the water bodies with shallow water depths, and MNDWI is not suitable for the index calculation of the whole image, and it is easy to produce large outliers that cannot be eliminated. This type of a traditional water body extraction method is focused on the definition of a threshold problem in the final analysis, although there are some improvement methods to enhance the water body recognition effect. Currently, most of the thresholds for water bodies are set manually, while different regions, different phases, and different satellite sensors may lead to different thresholds and bias in the water body recognition results, and the surrounding temperature and the physical environment also have a certain impact on the water body extraction [13], so the optimal thresholds are difficult to determine. Remote sensing images of different scenes often rely on subjective experience to set a threshold, and the threshold adjustment of the feedback image requires a large amount of a priori knowledge and even need experts; additionally, it cannot cope with a large amount of remote sensing image data using the traditional method of water body extraction, as it requires a large amount of manual threshold adjustment and remote sensing image software calculations, which is time-consuming and inefficient.

Remote sensing images of water body identification need infrared band information, and most of the high-resolution remote sensing images and unmanned aircraft remote sensing images lack the near-infrared band or short infrared band. In order to distinguish water bodies from vegetation, soil, etc. the visible light band within the range of 0.6 um is used, with less absorption and more projection. The reflectivity of clear water in the visible light band in the blue and green bands only reaches 4–5%, and in the red light band it is even only 2–3%, but the absorption is obvious in the range of the near-infrared band (NIR) to the short-wave infrared band (SWIR). High spatial-resolution satellite images have rich spatial and texture information, and researchers have constructed image classification and extraction methods by combining spectral, spatial, and texture features, which are capable of finer water body information extraction [1]. Manafd et al. [14] used eleven machine-learning methods for water body extraction in the northwest coast of Peninsular Malaysia. Typical machine learning methods such as support vector machines (SVM) [15] and random

forest [16] have poor generalization and instability problems that limit the generalizability of machine learning methods in water body extraction. With the explosion of hardware performance and the rapid growth of arithmetic power after the 21st century, Convolutional Neural Networks (CNNs) have developed at a high speed, and deep learning has been used in image classification, object recognition and classification, boundary segmentation and so on.

Currently, deep-learning water body extraction methods require a large number of original remote sensing images with labeled dichotomized images for feature learning and model modification, which need to be customized. The problems of feature recognition and classification and boundary delineation of remote sensing images are theoretically achievable by deep learning. Gang Fu et al. [17] used a variant on the basis of CNN, a fully convolutional network (FCN) model, on the true-color images of the Gaofen-2 satellite (GF-2) using 70 GF-2 images and tested using 4 other GF-2 images and 3 IKONOS true-color images to achieve deep learning for classification of remote sensing images. The method performed well in feature classification, but water bodies have the property of reflecting features. Thus, segmenting water bodies requires more complex deep learning models. Hu Wei [18] et al. used deep convolutional neural networks to classify hyperspectral images directly in the spectral domain, and the proposed method can achieve better classification performance than some traditional methods. The basic deep learning model still has unsatisfactory segmentation of water body boundaries although it has higher performance compared to traditional methods. Ordinary images do not have the large amount of spectral data that hyperspectral images have. Water bodies on ordinary images still need further segmentation methods. Feng et al. [19] used a deep U-Net network combined with conditional random fields (CRFs) to train the model on GF-2 images and WorldView2 images, and the effect was significantly improved compared to the basic model. The method is only for water body extraction for images in the near-infrared band. Shen et al. [20] introduced an attention mechanism into the U-Net and SegNet models and proved the new models' feasibility, but the accuracy improvement was not obvious. The self-attention mechanism is realized by filtering important information and filtering unimportant information, which leads to its ability to capture effective information being smaller than for CNNs. Thus, it leads to an insignificant increase in accuracy and unnecessarily increases the model parameters and training time.

Therefore, in this paper, we introduce a method based on a self-attention mechanism and the spatial pyramid pooling influence of a feature network, OCNet, in the U-Net network in order to improve the accuracy of extracting water bodies in remote sensing images and verify the effectiveness of the method. First, the original network is embedded with multi-sampling rate null convolution for the problem of fine water body recognition. Null convolution is able to control the effective sensing field [21] and deal with the large-scale variance of the object without introducing additional computation [22]. Second, to address the problem of reflecting surrounding features at the water body boundary leading to boundary blurring, the connection between the water body's parts as a whole is strengthened to capture global information, and a self-attention mechanism module is embedded in the network. The self-attention mechanism can expand the sensory field of convolutional computation and obtain more contextual information connections, which can improve the accuracy of semantic segmentation [23]. The main contributions of this paper are as follows:

1. Hollow convolution of water bodies improves the accuracy of fine water body recognition;
2. Introducing an attention mechanism to better delineate water body boundaries;
3. Proposing a composite model that substantially improves the recognition rate compared to the traditional threshold method and the base semantic segmentation model.

## 2. Materials and Methods

### 2.1. Introduction to the Dataset

The remote sensing image dataset used in this experiment is from Luo et al. [5], and is an automatic mapping dataset of surface water based on multispectral images. The remote sensing raw dataset is shown in Table A1 (Supplementary materials). The images were obtained from the Sentinel-2 satellite, a high-resolution multispectral imaging satellite consisting of Sentinel 2A and Sentinel 2B, which provides high-resolution images of vegetation, soil and water cover, inland waterways and coastal areas, making it particularly suitable for surface water monitoring. The MSI (multi-spectral imager) on Sentinel-2 is equipped with 13 operating bands, including blue (B2), green (B3), red (B4), and NIR (B8) in the 10 m band, red end (B5), NIR (B6, B7 and B8A) and SWIR (B11 and B12) in the 20 m band, and coastal (B11 and B12) in the 60 m band. The coastal atmospheric aerosol (B1) and cirrus (B10) bands are available in the 60 m resolution band.

For this experiment, 95 scenes from 31 Sentinel-2 images were extracted as the base dataset (as shown in Figure 1), including various surface water bodies such as reservoirs, rivers, lakes and oceans, as well as land cover types such as urban areas, roads, forests and glaciers. To delineate the boundary of water bodies, the surface water scenes were selected, and the uncertain parts were identified using Google Maps and other auxiliary data for scale. The water body parts and non-water body parts were binarized and labeled (as shown in Figure 2) to obtain labeled images. These labeled images were used for feature learning and model modification in the proposed OCNet method to improve the accuracy of water body extraction in remote sensing images.

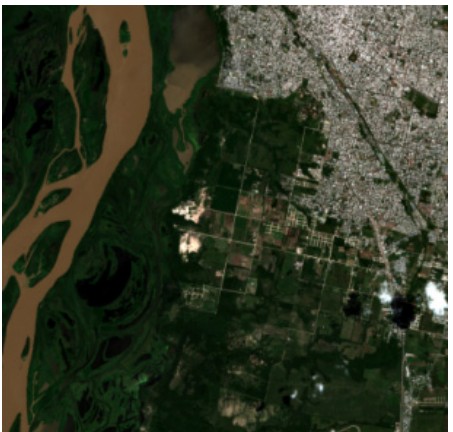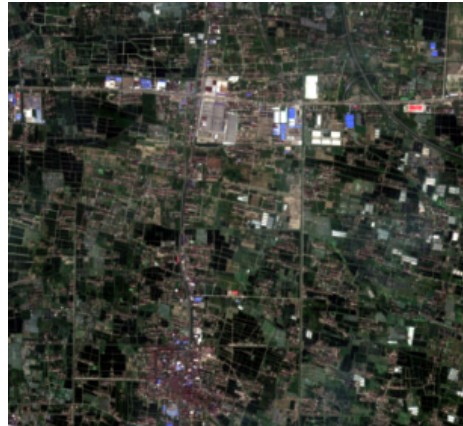

**Figure 1.** Partial sample of the original (RGB) image.

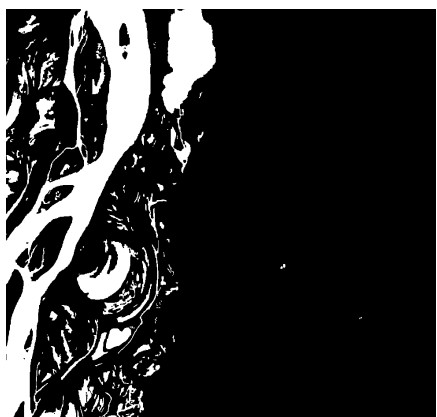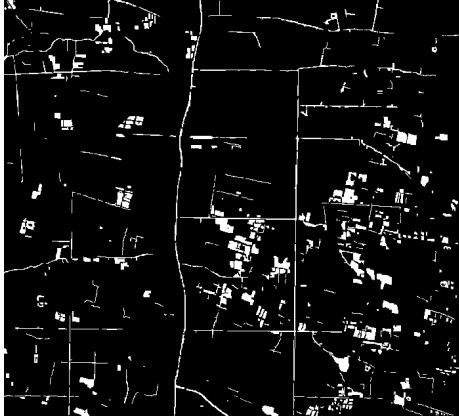

**Figure 2.** Label binarization (grayscale) partial sample.

### 2.2. Dataset Pre-Processing

To train the semantic segmentation models, the RGB bands (B4, B3 and B2) were extracted from the raw Sentinel-2 satellite images of the 95 scenes and synthesized into RGB images. As most of the current popular semantic segmentation models are trained using three-channel RGB images, the input image size of the deep learning model used in this paper was 256 × 256, and the RGB image and labeled binarized image were segmented into 256 × 256 × 3 and 256 × 256 × 1 preprocessed images, respectively, using Python image cropping. The preprocessing flowchart of the image dataset is shown in Figure 3.

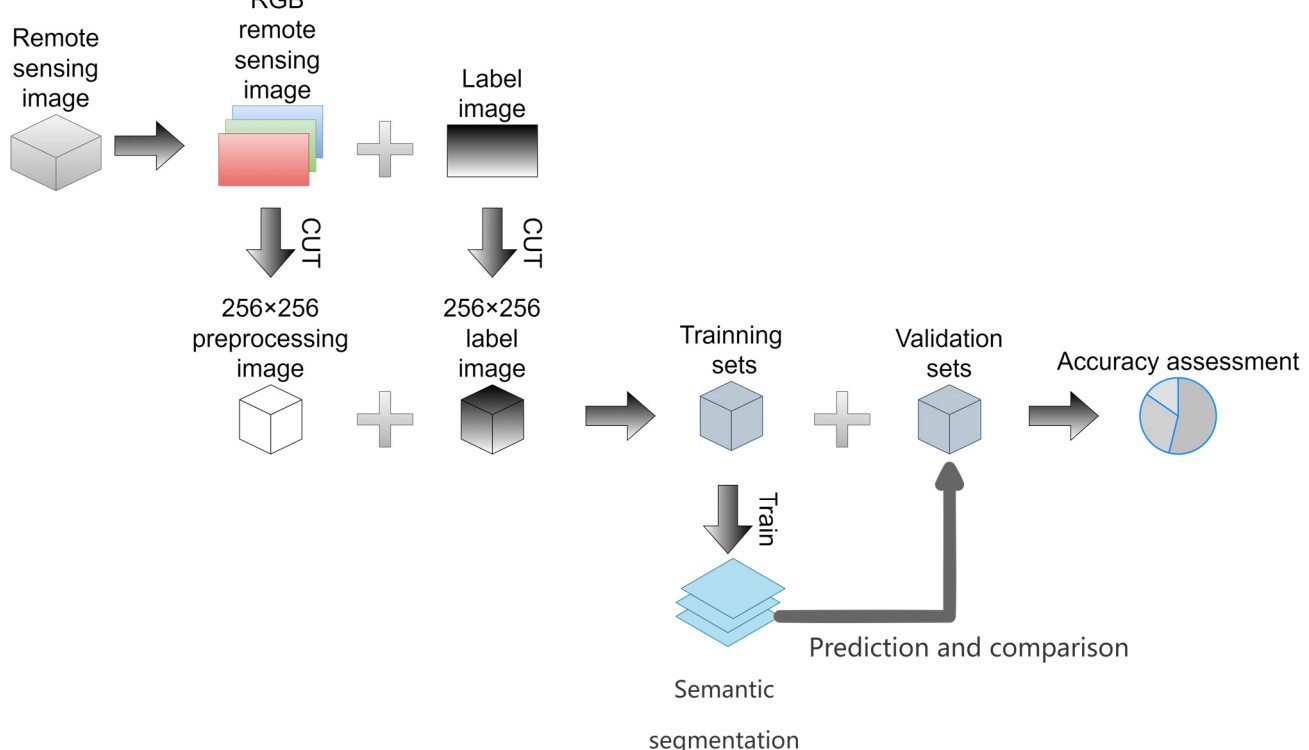

**Figure 3.** The process of dataset pre-processing.

After segmentation, a total of 1075 RGB preprocessed images were obtained, corresponding to 1075 labeled binarized images. Of these, 771 images from 64 scenes were used as the training dataset, and 304 images from 31 scenes were used as the validation dataset. The labeled images are grayscale images, where a water body value of 1 is represented as white, and a non-water body value of 0 is represented as black. The labeled samples corresponding to the preprocessed samples in Figure 4 are shown in Figure 5. These labeled images were used to train and validate the proposed OCNet method for water body extraction in remote sensing images.

### 2.3. General Module

In the experiment, the copy and crop operation in the U-Net model was replaced by the feature aggregation module to improve the accuracy of water body extraction in remote sensing images. The improved model, called OCNet, is shown in Figure 6. In the OCNet model, the atrous spatial pyramid pooling (ASPP) and self-attention operations are performed once on the feature maps after two 3 × 3 convolutions in each layer of the downsampling part. The obtained results are concatenated with the feature maps in the same layer of the upsampling part, while keeping the number of channels equal to the number of channels in the same layer of the original U-Net network. This helps to capture more contextual information and improve the accuracy of semantic segmentation.

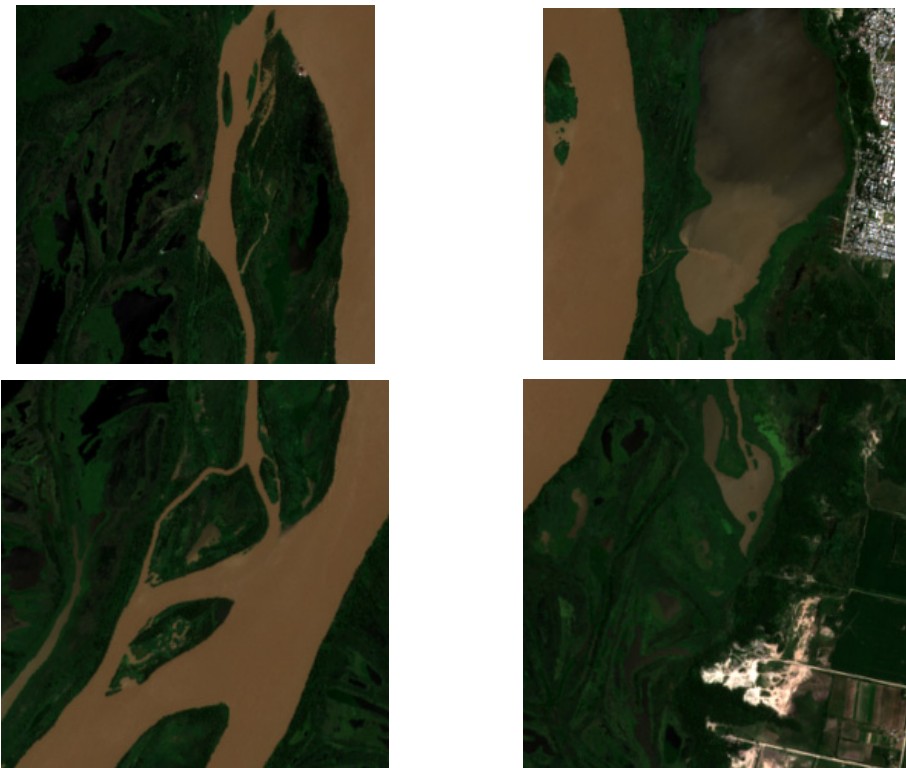

**Figure 4.** Sample of partially pre-processed images.

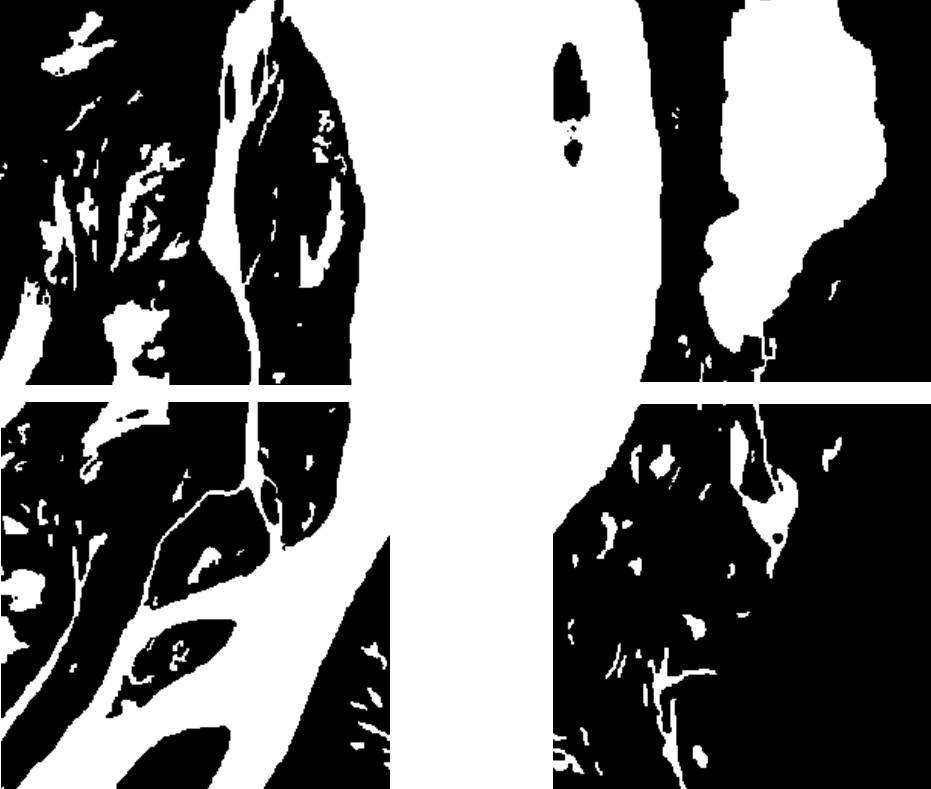

**Figure 5.** Sample of partially pre-processed labels.

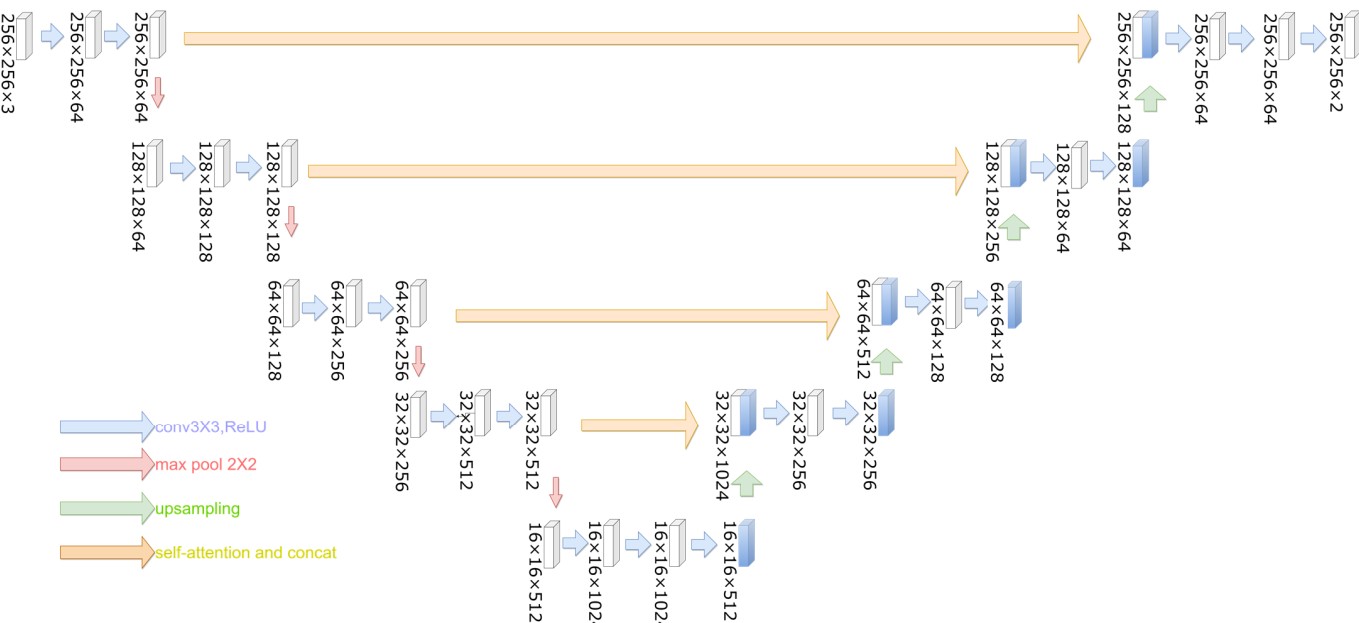

**Figure 6.** U-Net+OCNet model.

For a small number of remote sensing images with annotations, U-Net based architectures have proved to be an excellent choice for remote sensing image segmentation tasks; furthermore, the use of self-attentive mechanisms has been shown to provide high-capacity models that can appropriately utilize large-scale datasets [24].

The ASPP operation is used to enlarge the effective receptive field of the convolutional layer and capture multi-scale contextual information. The self-attention operation is used to enhance the feature maps by attending to the most relevant features in the same feature map. The feature aggregation module is used to capture the context information of the entire water body and reduce the blurring of the water body boundary caused by the reflection of surrounding features.

The proposed OCNet method was compared with threshold-based methods and the base semantic segmentation model. The experimental results show that the proposed method achieved better accuracy and efficiency in water body extraction from remote sensing images, demonstrating its potential in water resource management and environmental monitoring.

### 2.4. U-Net Model

The U-Net [25] network model was first proposed by Ronneberger et al. in 2015. Compared with traditional CNN models, U-Net can predict the classification of each pixel point in an image with a relatively small image dataset and accurately segment images into different classes. U-Net introduces a jump-join step, which can more accurately output the type of the image labels and achieve more accurate image segmentation. U-Net was initially applied to image segmentation in the biomedical direction, where the biomedical image background is more homogeneous and less complex, and the samples of the image dataset used are fewer, so good segmentation results can still be achieved by using a network model that has not much complexity or depth [26], and multispectral remotely sensed imagery that includes different types of features has a complex background and has multiple channels of data.

The U-Net model used in this paper is shown in Figure 7, and the main network architecture is divided into three parts: the decoder, bottleneck layer and encoder. The decoder, also known as the downsampling part, uses two $3 \times 3$ convolution kernels for all four layers with the activation function ReLU. It doubles the number of feature map channels (C) and uses a $2 \times 2$ maximum pooling layer with a step size of 2 after convolution.

The feature map width (W) and height (H) are reduced by half with each maximum pooling layer. The bottleneck layer consists of three 3 × 3 convolution kernels, keeping the bottom layer, which is the minimum feature map C, unchanged. The encoder, also known as the upsampling part, contains four blocks similar to the downsampling part. Each block is first upsampled by 2 × 2 with a step size of 2. Then, the feature map of the same level as the downsampling part, i.e., the same W and H, is subjected to a Concat operation. Two 3 × 3 convolution kernels with an activation function of ReLU are used to reduce the C value of the feature map. Finally, a 1 × 1 convolution operation is performed, and the number of convolution kernels for the binary classification segmentation task is 2 to obtain the output binarized labeled map.

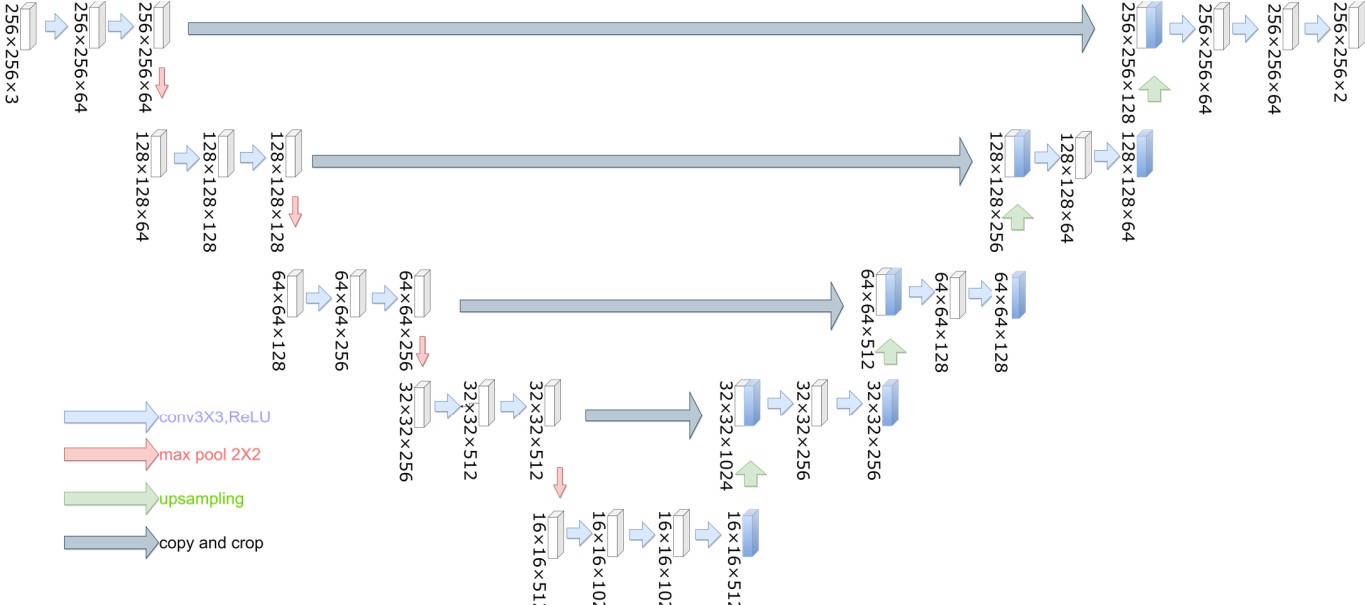

**Figure 7.** U-Net model.

The U-Net model is widely used in the field of image segmentation due to its ability to capture both local and global features. However, it has limitations in handling complex scenes, and the use of the jump-join step may cause information loss. Therefore, in this paper, an improved U-Net model, called OCNet, is proposed by introducing the ASPP and self-attention operations and replacing the copy and crop operation with the feature aggregation module. The experimental results show that the proposed OCNet method achieves better accuracy and efficiency in water body extraction from remote sensing images.

## 2.5. Self-Attention Mechanism

The self-attention mechanism was proposed by the Google team in the Transformer model in 2017 [27]. There is a certain relationship between different pixel points in the input image, and the traditional CNN network cannot effectively use the relationship between different pixel points in the actual training, leading to unsatisfactory results. The introduction of the self-attention mechanism can effectively improve this issue. The self-attention models perform significantly better than the convolutional baseline, and the addition of self-attentive mechanisms to convolutional networks can result in strong accuracy gains [23]. Figure 8 shows the basic structure of the self-attention mechanism. A layer of feature maps in the deep convolutional network is used as input, and the input maps are convolved three times with 1 × 1 kernels to obtain three feature maps of key, query and value. The three feature maps are then used to calculate the self-attention maps:

1.  In the first step, the key and query are multiplied by the tensor matrix to obtain the attention matrix.
2.  In the second step, the attention matrix is normalized using the softmax function to obtain the attention weight map.
3.  In the third step, the attention map and value are multiplied by the tensor matrix to obtain the final self-attention feature maps.

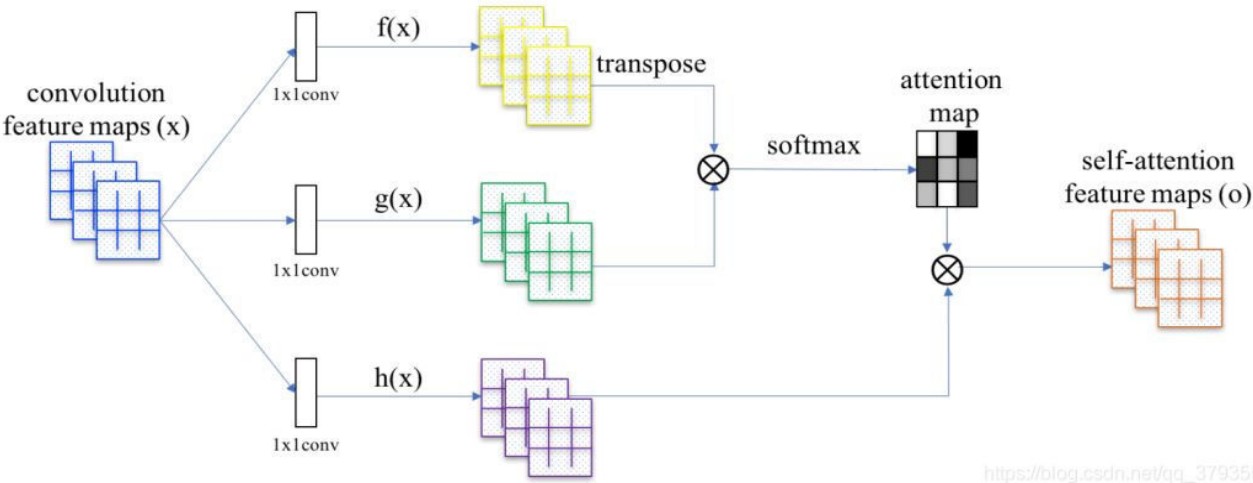

**Figure 8.** Self-attention mechanism.

The self-attention mechanism can effectively capture the relationship between different pixel points in the input image and highlight the important features, which can improve the accuracy of semantic segmentation. In this paper, the self-attention mechanism is introduced into the proposed OCNet method to enhance the feature maps and improve the accuracy of water body extraction in remote sensing images. The experimental results show that the proposed method achieves better accuracy and efficiency compared to the traditional U-Net model and threshold-based methods.

*2.6. ASPP Model*

Figure 9 shows the schematic diagram of the atrous spatial pyramid pooling (ASPP) model. The input feature maps are subjected to dilated convolution operations with different sampling rates to obtain feature maps with different sampling rates. The dilated convolution operation is a type of convolution that expands the receptive field without increasing the number of parameters. The feature maps with different sampling rates are then concatenated, and finally, the number of channels is reduced by the convolution of $1 \times 1$ kernels to obtain the expected value. The role of the ASPP module is to capture multi-scale features and gain the accuracy of semantic segmentation. Based on the characteristics of remote sensing images with many details and a complex background, the ASPP module can improve the accuracy and quality of remote sensing image classification and extraction, and extract features from remote sensing images more effectively and clearly [28].

The ASPP model has been widely used in various semantic segmentation tasks due to its ability to capture multi-scale information and improve the accuracy of semantic segmentation. In this paper, the ASPP operation is introduced into the proposed OCNet method to capture more contextual information and improve the accuracy of water body extraction in remote sensing images. The experimental results show that the proposed method achieves better accuracy and efficiency compared to the traditional U-Net model and threshold-based methods.

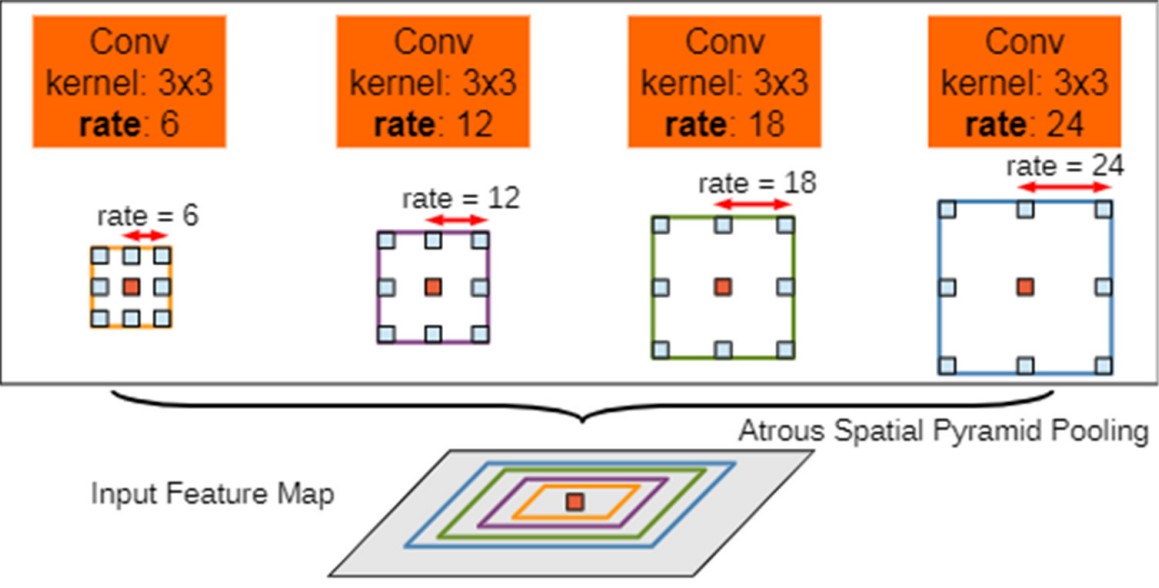

**Figure 9.** ASPP model.

*2.7. OCNet Model*

The OCNet [29] model was proposed by Yuan et al. in 2021 to achieve enhanced prediction of image information segmentation using inter-pixel semantic relationships. The proposed model includes a new layer of a feature aggregation model added after the traditional backbone network, focusing on efficiently modeling the dense relationships between pixels. Figure 10 shows a schematic diagram of the proposed OCNet model.

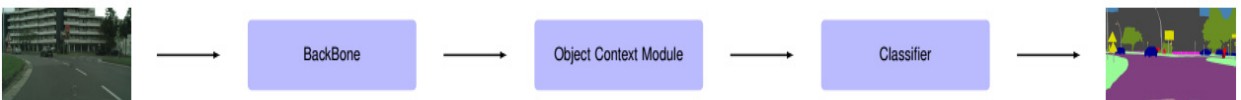

**Figure 10.** OCNet model.

The feature aggregation module is designed to capture the context information of the entire water body and reduce the blurring of the water body boundary caused by the reflection of surrounding features. The ASPP and self-attention operations are performed once on the feature maps after two $3 \times 3$ convolutions in each layer of the downsampling part. The obtained results are concatenated with the feature maps in the same layer of the upsampling part, while keeping the number of channels equal to the number of channels in the same layer of the original U-Net network. This helps to capture more contextual information and improve the accuracy of semantic segmentation.

In the experimental part of this study, the object context module was used to further improve the accuracy of water body extraction from remote sensing images. The object context module combines the OCNet model with the ASPP model, as shown in Figure 11.

The object context module consists of five convolution operations performed on the original feature maps to obtain five copies of feature maps with invariants H, W and C. Three of them are sampled with 12, 24 and 36 dilated convolutions, one with a $1 \times 1$ convolution, and one with a self-attention mechanism operation. The dilated convolutions with different sampling rates can capture multi-scale contextual information, while the $1 \times 1$ convolution and self-attention operation can enhance the feature maps and capture the inter-pixel semantic relationships. After the Concat operation on the five feature maps, a feature map with five times the number of channels is obtained. Finally, a $1 \times 1$ convolution is applied to reduce the number of channels to the original number of channels of the feature map.

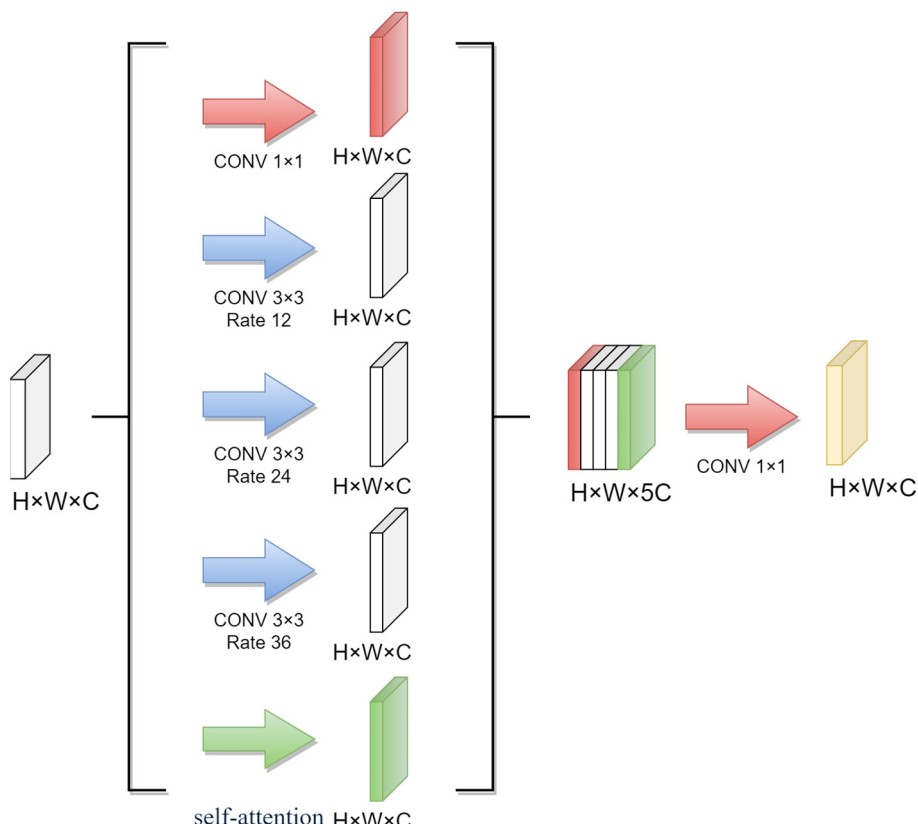

**Figure 11.** Object context module.

The object context module can effectively capture the context information and inter-pixel semantic relationships, which helps to improve the accuracy of water body extraction from remote sensing images. The experimental results show that the proposed object context module achieved better accuracy and efficiency compared to the traditional U-Net model, threshold-based methods, and the proposed OCNet method.

*2.8. Network Model Parameter Setting*

Due to the introduction of dilated convolution and a self-attention mechanism into the improved model proposed in this study, the computation time is increased due to the larger model size. To cope with this, the number of training epochs is set to five times, and the best model is selected from the five trained models. The increase in the number of parameters also leads to GPU memory tension, and therefore, the experimental data batch size [30] is set to 1. The batch size indicates the number of images per training, and the original U-Net network training can be adjusted to increase the batch size parameter to improve the training efficiency.

In the training process, the Adam optimizer [31] is used instead of the traditional stochastic gradient descent algorithm. The Adam optimizer can automatically adjust the learning rate (lr) with an initial value set to 0.0001 and the smoothing constants $\beta_1$ and $\beta_2$ set to 0.5 and 0.999, respectively. The accumulated gradient is calculated using Equation (1):

$$m_t = \beta_1 * m_{t-1} + (1 - \beta_1) * g_t \tag{1}$$

In Equation (1), t denotes the number of calculations and the calculated gradient. The square of the accumulated gradient is calculated as in Equation (2):

$$v_t = \beta_2 * v_{t-1} + (1 - \beta_2) * (g_t)^2 \tag{2}$$

The biases correction (m) is calculated as in Equation (3):

$$\hat{m}_t = \frac{m_t}{(1 - \beta_1)^t} \tag{3}$$

The biases correction (v) is calculated as in Equation (4):

$$\hat{v}_t = \frac{v_t}{(1 - \beta_2)^t} \tag{4}$$

The update parameter (θ) is calculated as in Equation (5):

$$\theta_t = \theta_{t-1} - \frac{\hat{m}_t}{\sqrt{\hat{v}_t} + \varepsilon} lr \tag{5}$$

The CrossEntropyLoss [32] function is used as the loss function in the training process. The CrossEntropyLoss() function actually takes the output and performs a sigmoid function [33], which sets the data to between 0 and 1, and then places the data into the traditional cross-entropy function to obtain the result. The formula for the cross-entropy loss function is given in Equation (6):

$$L = -[y * \log \hat{y}_t + (1 - y) * \log(1 - \hat{y}_t)] \tag{6}$$

where y_i is the ground truth label for pixel i, and p_i is the predicted probability of pixel i belonging to the positive class. The sigmoid function is given in Equation (7):

$$g(s) = \frac{1}{1 + e^{-s}} \tag{7}$$

The use of the Adam optimizer and CrossEntropyLoss function can help to improve the training efficiency and accuracy of the proposed model.

## 3. Results

### 3.1. Experiment Environment

The experiments in this paper were conducted on a Windows 10 system, and the hardware and software development environment configurations are shown in Tables 1 and 2, respectively.

**Table 1.** Hardware development environment configuration.

| Hardware | Model |
|---|---|
| CPU | AMD 3700X 8Core 3.6 GHz |
| GPU | NVIDIA RTX2070Super 8 GB |
| RAM | DDR4 32 × 2 GB |
| Storge | Nvme 1 TB |
| Power supply | 550 W |
| Motherboard | Asus B450m pro gaming |

**Table 2.** Software development environment configuration.

| Environment | Model |
|---|---|
| Development environment | Anaconda 3 |
| Programming language | Python 3.9.13 |
| Deep learning framework | Pytorch 1.13.0 |
| GPU drive | NVIDIA CUDA 11.7 |
| Compile environment | PyCharm Community Edition 2021.3.2 |
| Computer system | Windows 10 |

### 3.2. Evaluation Indicators

The performance evaluation of the deep learning model is a direct manifestation of the performance improvement achieved by the proposed method in this paper. Different evaluation metrics are used to extract the advantages and disadvantages of different aspects of the model using different algorithms. In this paper, five evaluation metrics were mainly used to evaluate the model in semantic segmentation experiments, which are accuracy (ACC), precision, recall, mean intersection over union (miou) and F1 score.

The accuracy (ACC) is calculated using Equation (8):

$$ACC = (TP + TN)/(TP + TN + FP + FN) \tag{8}$$

where TP indicates the number of pixels where the predicted image is classified as a water body and the labeled image is also a water body; TN indicates the number of pixels where the predicted image is classified as a non-water body and the labeled image is also a non-water body; FP indicates the number of pixels where the predicted image is classified as a water body and the labeled image is a non-water body; and FN indicates the number of pixels where the predicted image is classified as a non-water body and the labeled image is a water body.

The precision, recall and F1 score are calculated using Equations (9)–(11), respectively:

$$precision = TP/(TP + FP) \tag{9}$$

$$recall = TP/(TP + FN) \tag{10}$$

$$F1 = 2 * ((precision * recall)/(precision + recall)) \tag{11}$$

where TP, TN, FP and FN are defined as before.

The miou is a commonly used evaluation metric for semantic segmentation and is calculated as the mean intersection over union of all classes in the dataset. The miou is calculated using Equation (12):

$$miou = (TP/(TP + FP + FN) + TN/(TN + FN + FP))/2 \tag{12}$$

Mean intersection over union (MIoU) is a standard evaluation metric for semantic segmentation, which calculates the ratio of intersection and union of ground truth predicted segmentation.

The use of multiple evaluation metrics helps to comprehensively evaluate the performance of the proposed method in different aspects, and to compare it with other methods in terms of accuracy, precision, recall, miou and F1 score.

### 3.3. Analysis

In the experiment, a total of 771 images sized $256 \times 256 \times 3$ with 771 corresponding labeled images sized $256 \times 256 \times 1$ were used as the training set. The OCNet network was added to the four layers of the original U-Net. The optimal model obtained after training was used to predict 304 images sized $256 \times 256 \times 3$, and the specific evaluation data are shown in Table A2.

From the data analysis of Figure 12, it can be observed that applying the OCNet network to the shallow network of U-Net was not effective. Even with the participation of the OCNet module in the shallow network, the recall and F1 evaluation metrics decreased significantly compared to the original network from 0.3105 and 0.3634 to 0.0296 and 0.0553, respectively. Therefore, it can be concluded that the OCNet module is not suitable for extracting features on the shallow network of U-Net, and is even less accurate than the original network.

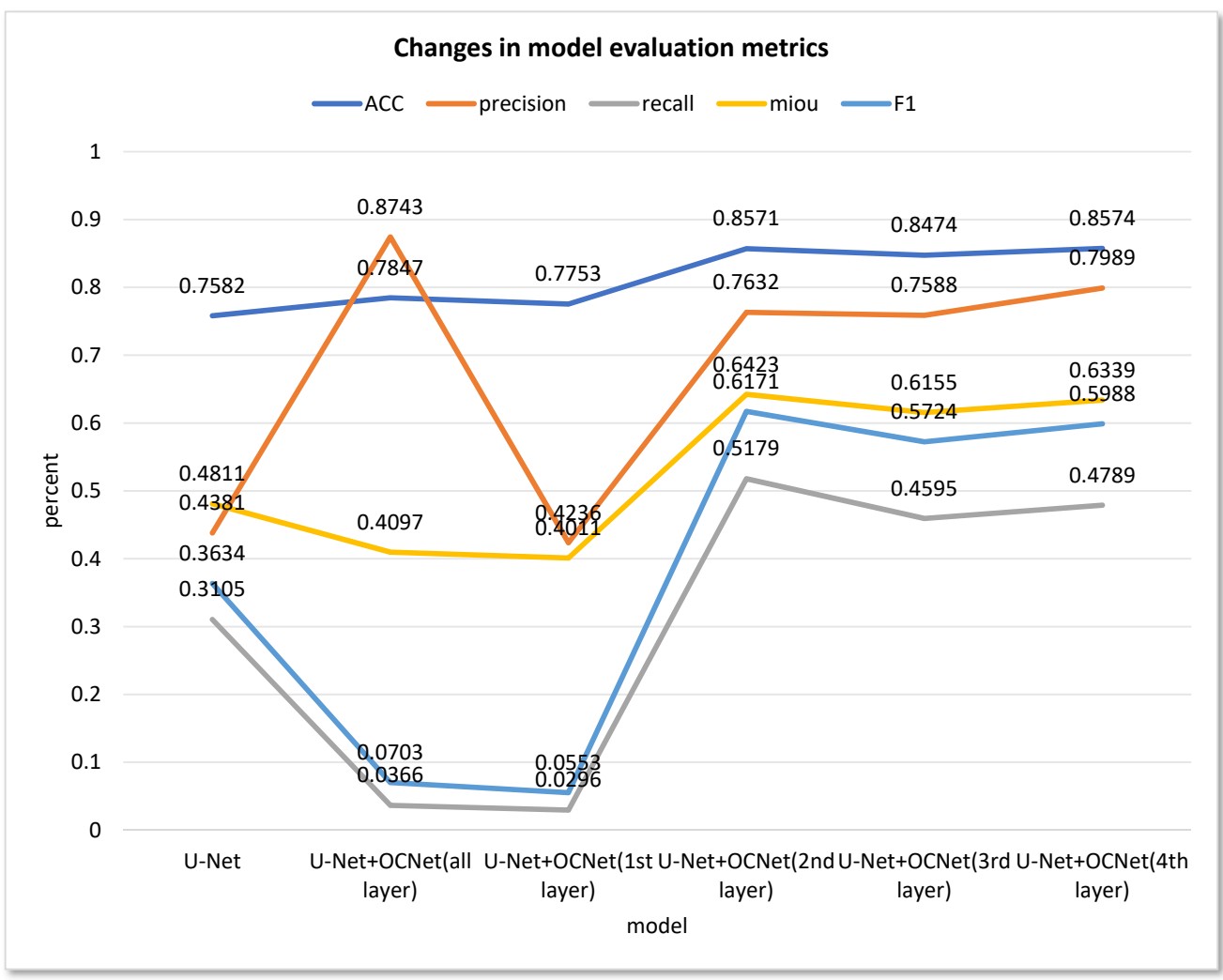

**Figure 12.** Changes in model evaluation metrics.

However, using the OCNet module in layers 2–4 of the U-Net network substantially improved the evaluation metrics obtained. The ACC increased from 0.7582 to about 0.85, the precision increased from 0.4381 to more than 0.75, and the three evaluation metrics of recall, miou and F1 were also substantially improved. Therefore, it can be concluded that replacing the Concat operation in the deep network of U-Net with the OCNet module is an effective way to improve the segmentation results.

In summary, the experimental results demonstrate the effectiveness of the proposed method in improving the accuracy of water body extraction from remote sensing images. The use of the OCNet module in layers 2–4 of the U-Net network can significantly improve the segmentation performance, while applying the OCNet network to U-Net's shallow network is less effective and reduces the accuracy while increasing the complexity of the network. Although the OCNet module can effectively improve the accuracy, it can only be fully utilized on the basic features extracted from the backbone network with a certain depth.

*3.4. Prediction Effect*

The predicted image results are compared in Figure 13 to demonstrate the effectiveness of the proposed method in improving the accuracy of water body extraction from remote sensing images. The results show that the proposed method exhibited superior adaptability to a range of remotely sensed images captured by any satellite, as well as RGB color images, compared to the original U-Net semantic segmentation approach. For both remote sensing

images and RGB color images, single- or multi-band images can be extracted and added to the OCNet network for deep learning in semantic segmentation projects.

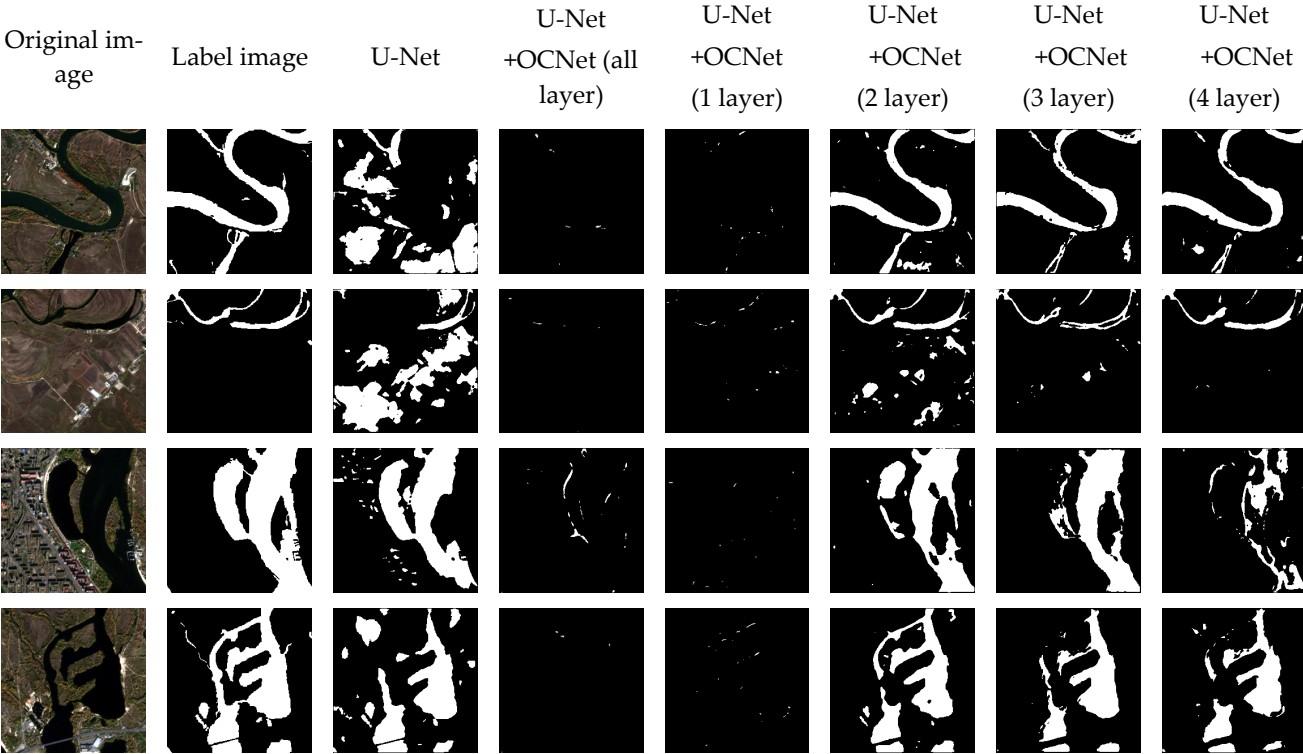

**Figure 13.** Comparison of prediction plots of 6 different models.

In particular, the proposed method demands a relatively small dataset, consisting of only 31 Sentinel-2 satellite images, and requires a comparatively shorter duration for model training and prediction in comparison to other machine learning and thresholding methods that necessitate large-scale image datasets and extensive manual segmentation to generate labeled maps of water bodies. Despite this, the proposed method achieved commendable segmentation outcomes for both water and non-water bodies, as evidenced by a high overall evaluation index. Therefore, the proposed method has the potential to be widely used in various applications related to water body extraction from remote sensing images. It can provide a more efficient and accurate solution for water resource management, environmental monitoring and other related fields. Compared to the traditional thresholding method, which requires manual adjustment of parameters such as NDWI according to different satellites, the deep learning approach can realize fast and convenient water body extraction by simply importing the required band images. Additionally, the improved deep learning method in this paper showed a substantial improvement over the original U-Net network in all indicators. For example, by adding OCNet to the fourth layer of the U-Net network, ACC was improved by 10%, precision was improved by 36%, recall was improved by 17%, miou was improved by 15% and F1 was improved by 24%.

## 4. Discussion

In the experiments of this paper, the step of replacing Concat in the U-Net network with the processed feature map of the OCNet model is proposed, which incorporates the ASPP module and the self-attention module compared to the original U-Net network. The ASPP module is able to expand the receptive field while maintaining the resolution, and also captures multi-scale contextual information with different ratios of null convolution kernels in order to facilitate the recognition of fine water bodies [34]. The self-attention mechanism enhances the model's ability to capture target context information, improves

the model's detection performance of densely distributed targets, and improves the water body boundary ambiguity problem [35]. In addition to the U-Net network, the choice of backbone network can be adjusted for different requirements of accuracy and training time. Deep networks such as ResNet, ResNeXt, AmoebaNet, etc., or lightweight MobileNet, ShuffleNet, SqueezeNet, Xception, MobileNetV2, etc., can be selected. In this paper, the OCNet module is added to the deep feature map for improvement, which improves the shortcomings of the traditional model such as low prediction accuracy and the need for a large-scale training set, etc. Li et al. [36] used the traditional thresholding method to extract the water bodies on the images of ZY-3, and the water bodies at the same location in different periods need to be repeatedly experimented with in order to set the thresholds. The semantic segmentation method can optimize the extraction of water bodies on remote sensing images to a certain extent, but the extraction consumes a large amount of manual labor, resulting in slow efficiency.

The single-band thresholding method [6] and the water body index method [7–9] both need to be combined with multi-band images to adjust the threshold to achieve water body extraction, and are only applicable to multi-band remote sensing images. The traditional thresholding method of the water body index is only applicable to multi-band remote sensing images, and most of the high resolution images and UAV remote sensing images are not available in the near-infrared and mid-infrared bands, which are not applicable to water body index thresholding for water body extraction [37]. The semantic segmentation model proposed in this paper extracts only three bands of RGB data and labeled maps for training and prediction of the original remote sensing images. Therefore the method application is not limited to satellite image data, on aerial images such as UAV remote sensing data. In this paper, the water body extraction method based on deep learning improves the problem of the lack of bands that cannot be applied to the water body index, expands the types of images that can be extracted from the water body, and provides a new method for the application of high-resolution imagery and UAV remote sensing imagery in the setting of the water body.

## 5. Conclusions

In this paper, a deep learning-based water body extraction method for remote sensing images is proposed, which effectively improves the shortcomings of traditional methods such as the threshold method, which is not intelligent, and the shortcomings of the traditional semantic segmentation model, which is prone to lose the details of remote sensing images leading to the decrease of the accuracy rate. Operations such as the pooling of pyramids (ASPP) and the mechanism of self-attention are added, which effectively improves the assessment index of the predicted water body labeling map. For remote sensing images taken by different satellites, only the data of the red, green and blue channels need to be extracted, and a small number of images for model training can achieve a good water body extraction effect, which effectively improves the generalization of a single semantic segmentation model applicable to different satellite datasets. It is believed that the method in this paper can play a certain positive role in the fields of water resources acquisition, feature composition analysis, and flood warning.

The results achieved using the improved network in this paper show a substantial improvement in each of the evaluation metrics over the results achieved by the U-Net network. Incorporating the OCNet model into the deep network of the U-Net network, the experimental results of the various evaluation indexes have different degrees of improvement. Among them, ACC is improved by 10%, precision by more than 30%, recall by more than 15%, miou by more than 13% and F1 by more than 20%. For images from different satellites, the prediction accuracy of the same model has a large gap; Shen et al. [20] used the U-Net network and the improved S&CMNet network based on U-Net on remote sensing images captured by GF-6 with a small gap in the prediction results, and the method in this paper achieves more improvement, which verifies the feasibility of the improved basic network of OCNet.

**Supplementary Materials:** Dataset: https://zenodo.org/record/5205674 (accessed on: 1 January 2023), The following supporting information can be downloaded at: https://github.com/weng47846 2852/water-exaction (accessed on: 1 January 2023).

**Author Contributions:** Y.W. (Yijie Weng): conceptualization, methodology and writing; Z.L.: writing—review and editing; G.T.: software; Y.W. (Yang Wang): editing. All authors have read and agreed to the published version of the manuscript.

**Funding:** This research was funded by the National Natural Science Foundation of China, grant number 41501448, Xiamen University Technology, grant number XPDKT20029, Xiamen Municipal Bureau of Ocean Development, grant number 18CFW030HJ10, Natural Science Foundation of Fujian Province of China, grant number 2021H0026, Natural Science Foundation of Xiamen, grant number 3502Z20227067.

**Data Availability Statement:** Not applicable.

**Acknowledgments:** The authors would like to thank my supervisor, Zongmei Li from Xiamen Institute of Technology, for her detailed guidance on each part of the study. I also thank the Sentinel-2 satellite for providing the remote sensing dataset and Xin Luo et al. for compiling the dataset (https://zenodo.org/record/5205674 (accessed on: 1 January 2023)).

**Conflicts of Interest:** The authors declare no conflict of interest.

## Appendix A

**Table A1.** Remote sensing raw dataset list.

| ID | Image |
| --- | --- |
| 01 | S2A_MSIL2A_20190125T062131_N0211_R034 |
| 02 | S2A_MSIL2A_20190206T140051_N0211_R067 |
| 03 | S2A_MSIL2A_20190314T104021_N0211_R008 |
| 04 | S2A_MSIL2A_20190716T065631_N0213_R063 |
| 05 | S2A_MSIL2A_20190811T185921_N0213_R013 |
| 06 | S2A_MSIL2A_20190817T023551_N0213_R089 |
| 07 | S2A_MSIL2A_20190830T042701_N0213_R133 |
| 08 | S2B_MSIL2A_20181226T004659_N0211_R102 |
| 09 | S2B_MSIL2A_20190225T013649_N0211_R117 |
| 10 | S2B_MSIL2A_20190430T075619_N0211_R035 |
| 11 | S2B_MSIL2A_20190506T163849_N0212_R126 |
| 12 | S2B_MSIL2A_20190607T171909_N0212_R012 |
| 13 | S2B_MSIL2A_20190620T140739_N0212_R053 |
| 14 | S2B_MSIL2A_20190801T180929_N0213_R084 |
| 15 | S2B_MSIL2A_20190807T032549_N0213_R018 |
| 16 | S2B_MSIL2A_20190811T095039_N0213_R079 |
| 17 | S2B_MSIL2A_20190818T075619_N0213_R035 |
| 18 | S2B_MSIL2A_20190904T024549_N0213_R132 |
| 19 | S2B_MSIL2A_20190912T002609_N0213_R102 |
| 20 | S2B_MSIL2A_20191023T082009_N0213_R121 |
| 21 | S2B_MSIL2A_20190831T145739_N0213_R082 |
| 22 | S2B_MSIL2A_20190303T102019_N0211_R065 |
| 23 | S2A_MSIL2A_20190318T033531_N0211_R061 |
| 24 | S2A_MSIL2A_20190426T142801_N0211_R053 |
| 25 | S2A_MSIL2A_20190429T143751_N0211_R096 |
| 26 | S2A_MSIL2A_20190508T080611_N0212_R078 |
| 27 | S2A_MSIL2A_20190518T001111_N0212_R073 |
| 28 | S2B_MSIL2A_20190620T040549_N0212_R047 |
| 29 | S2A_MSIL2A_20190724T043711_N0213_R033 |
| 30 | S2A_MSIL2A_20190725T154911_N0213_R054 |
| 31 | S2B_MSIL2A_20191015T085929_N0213_R007 |

**Table A2.** Evaluation data.

| Model (Epoch 5) | ACC | Precision | Recall | Miou | F1 |
|---|---|---|---|---|---|
| U-Net | 0.7582 | 0.4381 | 0.3105 | 0.4811 | 0.3634 |
| U-Net+OCNet(all layer) | 0.7847 | 0.8743 | 0.0366 | 0.4097 | 0.0703 |
| U-Net+OCNet(1 layer) | 0.7753 | 0.4236 | 0.0296 | 0.4011 | 0.0553 |
| U-Net+OCNet(2 layer) | 0.8571 | 0.7632 | 0.5179 | 0.6423 | 0.6171 |
| U-Net+OCNet(3 layer) | 0.8474 | 0.7588 | 0.4595 | 0.6155 | 0.5724 |
| U-Net+OCNet(4 layer) | 0.8574 | 0.7989 | 0.4789 | 0.6339 | 0.5988 |

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
