# Peer review of "OCNet-Based Water Body Extraction from Remote Sensing Images"

_water, doi:10.3390/w15203557_

Round 1

Reviewer 1 Report

1.          What is the meaning of Y axial in Figure 12?

2.          Why the performance of using all layer is worse than use 1 to 4 layers in figure 12.

3.          The (a) and (b) should combined together for easy comparison in figure 13. The new method mentions that it can provide a more efficient solution for water resource management in line 469. So, how efficient?

4.          The reference format is not consistent. Some has “pp.” and some are not.

5.          What are the results of this paper?

Author Response

We deeply thank you for your insightful comments and suggestions to this manuscript. The manuscript has been substantially revised following your comments. The revisions were marked in red font. We also responded point by point to each comment as listed below.

Reviewer 2 Report

The authors present an interesting study on using a modified OCNet model for automatic water body extraction from remote sensing imagery. The proposed network architecture for deep learning seems well-designed, and the initial results appear promising. However, some minor revisions could help improve the clarity.

- If possible, consider expanding the background motivation in the introduction to highlight the specific limitations of prior remote sensing and deep learning methods that this new approach aims to address.

- The reported 85% accuracy of OCNet is noteworthy. It would be helpful to provide some context on how this compares with previous benchmarks in the field.

- The figures could be improved by increasing font sizes and legend clarity for better readability. If possible, it would be best to keep the figure on a single page rather than splitting it across multiple pages. This will help ensure the figure is viewed as a cohesive unit and improve readability.

- Table 1 may be missing some information based on the caption - please double check that the content is complete.

- The article would benefit from some language polishing for clarity and concision. Some sentences could be shortened or split into two for better flow. Carefully proofread to fix any minor grammar and formatting issues.

This study is well executed and makes a valuable contribution, pending some minor revisions to further improve clarity, context, and concision. I believe the authors can address these revisions in a straightforward manner. I look forward to seeing the revised manuscript and recommend publication after minor revisions.

Author Response

(The authors gave the same response as above.)

Reviewer 3 Report

The work presented to me for evaluation is interesting and has publication potential, but it would be necessary to specify what is new in relation to previous works. The weak element of the work is a rather laconic review of the literature on the discussed topic.

Author Response

(The authors gave the same response as above.)

Round 2

Reviewer 1 Report

1.          The format and font of figure 12 does not consist with the contents. Please redraw or revise it.

2.          The response says to merge 13(a) and 13(b), but it still has (a) and (b) in Figure 13?

3.          In line 469, says this study can provide a more efficient and accurate solution for water resource management, environmental monitoring, and other related fields. Please presents the value of efficient and improve percentage accurate.

4.          In Figure 7, some part is out of range?

Author Response

Thank you for providing your valuable comments again, we have revised the article accordingly as you suggested.
